# Regulatory and Policy-Related Aspects of Calcium Fortification of Foods. Implications for Implementing National Strategies of Calcium Fortification

**DOI:** 10.3390/nu12041022

**Published:** 2020-04-08

**Authors:** Gabriela Cormick, Ana Pilar Betrán, Fletcher Metz, Cristina Palacios, Filiberto Beltrán-Velazquez, María de las Nieves García-Casal, Juan Pablo Peña-Rosas, G. Justus Hofmeyr, José M. Belizán

**Affiliations:** 1Department of Mother and Child Health Research, Institute for Clinical Effectiveness and Health Policy (IECS-CONICET), Ciudad de Buenos Aires 1414, Argentina; belizanj@gmail.com; 2Departamento de Salud, Universidad Nacional de La Matanza (UNLAM), San Justo 1903, Argentina; 3UNDP/UNFPA/UNICEF/WHO/World Bank Special Programme of Research, Development and Research Training in Human Reproduction (HRP), Department of Sexual and Reproductive Health and Research, World Health Organization, 1211 Geneva 27, Switzerland; betrana@who.int; 4Department of Biology, Carleton College, Northfield, MN 55057, USA; metzf@carleton.edu; 5Department of Dietetics and Nutrition, Stempel School of Public Health, Florida International University; Miami, FL 33199, USA; crpalaci@fiu.edu; 6Department of Nutrition and Food Safety, World Health Organization, 1211 Geneva 27, Switzerland; beltranvelazquezf@who.int (F.B.-V.); garciacasalm@who.int (M.d.l.N.G.-C.); penarosasj@who.int (J.P.P.-R.); 7Effective Care Research Unit, Eastern Cape Department of Health, Universities of the Witwatersrand and Fort Hare, East London 5200, South Africa; justhof@gmail.com; 8Obstetrics and Gynaecology Department, University of Botswana, Private Bag UB 0022, Gaborone, Botswana

**Keywords:** calcium intake, inadequate intake, regulations, fortification, programs, LMICs

## Abstract

Daily calcium intake is well below current recommendations in most low- and middle-income countries (LMICs). Calcium intake is usually related to bone health, however an adequate calcium intake has also been shown to reduce hypertensive disorders of pregnancy, lower blood pressure and cholesterol values, and to prevent recurrent colorectal adenomas. Fortification of foods has been identified as a cost-effective strategy to overcome micronutrient gaps in public health. This review summarizes regulatory aspects of fortification of commonly consumed foods with micronutrients, with an emphasis on calcium. We selected a convenient sample of 15 countries from different WHO regions and described the regulatory framework related to calcium fortification of staple foods. We assessed the relevant policies in electronic databases including the WHO Global database on the Implementation of Nutrition Action (GINA) for fortification policies and the Global Fortification Data Exchange Database, a fortification database developed and maintained by Food Fortification Initiative (FFI), Global Alliance for Improved Nutrition (GAIN), Iodine Global Network (IGN), and Micronutrient Forum. Food fortification with micronutrients is widely used in many of the selected countries. Most countries had national legislation for the addition of micronutrients to staple foods, especially wheat flour. These national legislations, that includes regulations and standards, can provide the framework to consider the implementation of adding calcium to the fortification strategies, including the selection of the adequate food vehicle to reach the targeted population at risk safely. The strategy to include calcium in the fortification mix in fortified staple foods seems promising in these countries. However, potential undesired changes on the organoleptic characteristics of fortified foods and products thereof, and operational feasibility at the manufacturing site should be evaluated by the stakeholders during the planning stage. Codex Alimentarius standards should be considered by regulators in order to assure adherence to international standards. While the selected countries already have established national regulations and/or standards for fortification of key staple food vehicles, and there are experiences in the implementation of fortification of some staple foods, national food intake surveys can help plan, design, and modify existing fortification programs as well as monitor food and nutrient consumption to assess risk and benefits.

## 1. Introduction

Daily calcium intake is below recommendations in many low- and middle-income countries (LMICs), existing important inequities with high-income countries [1,2]. Low calcium intake is also observed in particular age groups such as adolescents, older adults, and those with a lower income from high-income countries [3,4,5,6,7,8,9]. The risk of inadequate calcium intake is consistently high in many countries in Asia, Africa, and Latin America. Patterns of calcium intake are variable in different populations. One study, based on adjustments and calculations from the Food and Agriculture Organization of the United Nations (FAO)’s national food balance sheets from 1992 to 2011 reported per-capita calcium supply (a proxy for consumption) by world regions of 474 ± 188 mg/day in Africa, 639 ± 223 mg/day in Asia, 858 ± 234 mg/day in the Americas, 982 ± 130 mg/day in Europe, and 936 ± 50 mg/day in Oceania, representing a risk of calcium deficiency of 80 ± 31, 57 ± 36 29 ± 27, 11 ± 7 and 11 ± 4%, respectively [10]. A review reporting calcium dietary intake found that the average national dietary calcium intake ranges from 175 to 1233 mg/day (78 studies from 74 countries) however many countries in Asia have an average dietary calcium intake of less than 500 mg/day [2]. Countries in Africa and South America present calcium intakes of between about 400 and 700 mg/day, although there were many countries with no information that would probably reduce these averages [2]. A review reporting the dietary intake of pregnant women shows consistently low calcium intakes across Asian, African, and Latin American countries (105 studies, 73,958 pregnant women from 37 countries) [1]. The mean calcium intake in pregnant women of low- and middle-income countries was 648 mg/day (95% confidence interval (CI) 569–727) whereas that from high-income countries was 948 mg/day (95% CI 872–1024) [1].

Inadequate calcium intake has usually been related to bone health problems. Improving calcium intake has also shown to add many benefits to health such as reduction of pregnancy-related hypertensive disorders, lower blood pressure particularly among young people, prevention of recurrent colorectal adenomas, lower cholesterol values, and fewer cases of high blood pressure among the offspring of women supplemented during pregnancy [11,12,13,14,15].

Considerations regarding the interpretation of the results of studies involving calcium supplements or increasing dietary calcium intake from food sources need to be taken into account in this discussion. Furthermore, the value of results coming from randomized controlled trials and observational studies need to be considered. In a recent review of randomized controlled trials the common concern of potential adverse effects of increasing calcium intake like a detriment of iron status, formation of renal stones, and myocardial infarction in older people were challenged [16]. Nonetheless, in supplementation studies, some of these effects, such as post-partum bone resorption and gastrointestinal discomfort have been reported [16].

Adaptation to low calcium intakes has been described in balance studies [17]. However, adaptations to situations of low calcium intake involve a rise in PTH, which leads to an increase of intracellular calcium in vascular smooth muscles and adipocytes resulting in vasoconstriction and lipogenesis [18,19].

There are three broad approaches to improve micronutrient intake, that can be part of integrated approaches to address the nutrient gaps according to the specific vitamins and/or minerals. Behavioral interventions that ideal, though it relies on personal habits and ability which are difficult to change in the short term. Targeted or universal supplementation for individuals have also been used. Fortification or enrichment of foods or beverages, that are consumed by the population of interest can improve dietary intake of the micronutrient without major changes in cultural practices [20]. Recommendations to improve dietary calcium intake by increasing the consumption of calcium-rich foods and/or taking calcium supplements has been around for many years, however, these recommendations show little impact in LMICs. In countries where inadequate calcium intake is widespread, the adherence to a recommendation for oral calcium supplements has been a limitation [12]. Supplementation adherence is usually better in individuals that already have better micronutrient intake, leading to an improvement of the population diet but not necessarily for those more at need [21].

Food fortification has been identified as the most cost-effective strategy to overcome micronutrient deficiencies [22]. An analysis in the UK assessed that if calcium fortification prevented 2% of fractures annually, stopping mandatory calcium fortification would increase social care costs by £3.06 million per year and by £22.39 million per year in the National Health Service [23]. Sandman, in 2015, estimated that the annual costs for 200 mg calcium per day was €0.22 per person in 2014 and that the implementation of a vitamin D plus calcium fortification program in Germany would cost €41 million per year while saving €315 million per year as a result of reduced fracture costs [24].

Food fortification has the advantage of not requiring modification of food habits which can lead to a higher level of acceptability than other strategies [25]. Currently, more than 130 countries have mandatory fortification of salt with iodine, and around 85 have mandatory fortification of wheat flour with micronutrients such as iron, iodine, folate, and vitamin A [26] Food fortification strategies have been implemented for more than 80 years and have contributed to an improvement in health by lowering the incidence of goitre, anaemia, night blindness, rickets, beriberi, and pellagra [27,28]. Since micronutrients also have a long-term impact in achieving and maintaining optimum health across all life stages, food fortification strategies have moved from preventing the occurrence of these diseases to ensuring populations reach an adequate nutrient intake [8].

Fortifying foods with a micronutrient, for example calcium, could have important population-based benefits when [29]: Most of the individuals have an intake below the recommended or desirable level of calcium;The identified food fortified with the calcium is widely consumed in that population;Fortifying with the calcium will not likely create an imbalance of other nutrients (i.e. iron);Calcium added to a particular food is stable within the specified conditions of storage and use of that food;The bioavailability (efficiency of absorption) of calcium in the food matrix has been tested and it is desirable;There is low risk of toxicity from consumption of that food;Calcum-fortified items are available and accepted by all populations or populations at risk of deficiency.

A regulatory framework of food fortification is necessary to assure quality, safety, and efficacy of public health interventions [30]. A national food fortification program may be governed by a general legislation framework and through technical specifications that are included in standards and regulations, which are the most common legislative instruments applied to staple foods. Regulations and standards for the fortification of a staple food (vehicle) might include considerations on the recommended nutrients, target fortification levels, minimum and maximum levels, and fortification compounds; provisions on labelling, claims, and advertising; procedures for regulatory monitoring and product sampling; and enforcement measures to assure compliance. 

Other specifications included in the regulations and standards might incorporate physical, microbiological and contaminant properties. Several standards of the Codex Alimentarius provide general guidance regarding the addition of essential nutrients; labelling and claims; composition and quality; and food safety factors in staple grains [31,32,33,34].

National competent authorities should determine whether the addition of micronutrients (i.e. calcium) to staple foods should be mandatory or voluntary. Mandatory food fortification is usually recommended when micronutrient deficiencies are widespread in the population. Mandatory food fortification is called mass fortification when the objective is to reach the entire population or targeted fortification when it is intended for a special group, for example through a food program. For mandatory food fortification to be effective, food production needs to be industrialized and centralized [26]. If food production is spread among small manufacturers or the population consumes its own production, a lower impact of mandatory food fortification will be expected. Legislation mandating large-scale fortification of a staple food enables regulatory authorities to monitor fortification for compliance with their standards and this helps ensure quality. It also creates a business environment where all producers must fortify and therefore incur similar costs which helps secure the sustainability of the program [35,36]. On the other hand, voluntary food fortification is market-driven and the industry can opt to fortify foods in order to restore nutrients that may be lost during food manufacturing, storage, handling, or processing; or to improve the nutritional value of a food with the objective to increase sales [8,25]. In some countries where consumers’ interest in fortified foods is high, voluntary food fortification can have an impact similar to mandatory food fortification [27].

Recommended nutrients and the target fortification levels should be adapted according to the epidemiological needs of the country [27]. A wide variety of calcium compounds are currently used as food fortificants. Regulations and standards for fortification of staple foods can either include the list of all the recommended calcium-fortification compounds, or it can permit the use of specific compounds. In the development or modification of regulations and standards at country level should be included the recommended calcium compounds, added at concentrations necessary to achieve a satisfactory impact.

Once the recommended calcium compounds to be added are estimated, the design of the standard should consider the calcium content in the selected compound(s), the expected losses, and the intrinsic content in the staple food chosen, when defining the target fortification level for the staple food and this amount can be taken as a reference for internal quality control at the production sites. Afterwards, the allowable range of variation around the mean can be estimated for external quality control purposes [27,37].

This review summarizes and analyses some regulatory and policy-related aspects concerning the fortification of some staple foods with the aim to inform the planning and design of a calcium fortification strategy in different countries, in the case when this policy decision was decided as the best approach to reduce the calcium intake gap.

## 2. Materials and Methods

We selected a convenience sample of 15 countries from different WHO regions and described available fortification regulations of staple foods with an emphasis on calcium (see Table 1 for the list of the 15 countries included in this review). We conducted an search on electronic databases on calcium fortification, specifically in these countries. We also searched the Global database on the Implementation of Nutrition Action (GINA) for fortification policies and the Global Fortification Data Exchange Database, a fortification database developed and maintained by Food Fortification Initiative (FFI), Global Alliance for Improved Nutrition (GAIN), Iodine Global Network (IGN), and Micronutrient Forum. Additionally we contacted key stakeholders including the Food and Agriculture Organization and the World Health Organization for additional references. We focused our search on the following information:regulations or standards for fortification of staple foods with any micronutrient and specifically for calcium fortification, either mandatory or voluntary. Where regulations or standards are founded we looked for information on the percentage of food currently being fortified and on the impact that these regulations have on population dietary intakes;percentage of industrialized staple food at country level, as countries with low industrialized staple foods would limit the effectiveness of a national fortification program;estimates of staple food consumed at country level to assess the impact that a new fortification strategy could have on calcium intake.

## 3. Results

### 3.1. Summary of Regulations for Calcium Fortification of Staple Foods in Selected Countries.

The list of countries with a summary of their regulations for calcium fortification of staple foods is shown in Table 1.

Only one country (UK)has mandatory calcium fortification (i.e. wheat flour for baking), nine countries have voluntary fortification of calcium and six have no regulation for calcium fortification. The most common vehicles are grains such as wheat, rice, and maize and their products. The levels are typically expressed as mg of calcium per serving, ranging from 200 to 1000 mg of calcium per serving and as a proportion of calcium ranging from 94 to 390 mg of calcium per 100 grams of product.

### 3.2. Micronutrient Fortification Standards and Level of Grain Flour Available for Human Consumption

The Global Fortification Data Exchange provides information on current fortification standards, industrialized food production, food availability for human consumption, and percentage of some fortified foods (i.e. maize flour, oil, salt, rice, and/or wheat flour) [61]. The UK is the only country with mandatory fortification of wheat flour with calcium and has been since 1942. In that country it was estimated that, as 2018, 98% of wheat flour is industrially milled, 272 g of wheat flour is available for consumption per person per day, and 88% of wheat flour currently produced is being fortified (see Table 2). However, other countries have mandatory or voluntary regulations for the fortification of other staple foods with at least one micronutrient. Countries with extant regulations present an opportunity to add other micronutrients such as calcium to current regulations, after feasibility, organoleptic, and acceptance studies have been performed. In our convenience sample, countries have had regulations since 1995 in the USA to 2018 in India. Information on the percentage of flour industrially milled is important because industrially produced foods facilitate the attainment of mass fortification. For example, some countries such as Ethiopia, India, and Mozambique have a low percentage consumption of industrialized flour which would impair the implementation and lower the effectiveness of mandatory fortification of staple grains. In our convenience sample of countries, the percentage of flour industrially milled ranges from 0% in Tanzania to 100% in several countries. Ideally, data on dietary micronutrient intake from population nutritional surveys would be necessary to design a fortification strategy, however this information is not always available. On the other hand, daily food availability data that, although not the optimal measurement, is available for many countries. Daily food availability data is the amount of food available per individual in each country calculated from supplies through domestic production or imports, which does not consider in-house distribution of the fortified food. Many of the countries selected in this study have regulations for the micronutrient fortification of staple foods, especially flour, which is one of the most common vehicles for food fortification. Table 2 shows the daily flour availability for each of the selected countries and this information can be used to compare the amounts of flour available in different countries. In our convenience sample of countries, daily flour availability ranges from 375 grams/per capita/day in Jordan to 33 grams/per capita/day in Zambia. Finally, the reported percentage of food actually fortified provides an estimate of the success in the process of each fortification strategy. The percentage of flour actually fortified, set out in column 6 of Table 2 provides the percentage of food being fortified in each country. This number does not consider in-house distribution of the fortified food, and assumes that the consumption is equally distributed among all members of the household. In our convenience sample of countries, the percentage of food being fortified ranges from 0% in Ethiopia and Tanzania to 100% in Argentina and Canada. This indicates that for some countries, voluntary flour fortification achieves 100% fortification whereas in other countries, voluntary flour fortification leads to little impact. On the other hand mandatory food fortification does not always ensure 100% of the food being fortified. Voluntary legislation in China, Zambia, Tanzania, or Ethiopia show little or no impact on increasing the percentage of some fortified flour (0% to 54%), whereas in Jordan, United Arab Emirates and Qatar voluntary fortification seems to achieve very similar percentages of food fortification to a mandatory fortification strategy, above 90%.

Although China has voluntary rice, maize, and flour fortification and Denmark has voluntary flour fortification policies, this data is missing in the Global Fortification Data Exchange database. The Global Fortification Data Exchange database also provides data on USA mandatory flour fortification, however in the USA, flour fortification is non-mandatory [63]. For this reason, we excluded the information from the table. Although food fortification is voluntary in the USA, when a food manufacturer opts to label a certain food as enriched, it is mandatory to add some other micronutrients, and this is referred as to mandatory food fortification in the USA. In the United States, mandatory fortification (also referred as enrichment) is not mandated in the same ways as other countries, products formulated to conform to the standard of identity promulgated by the Food and Drug Administration (FDA) for the enriched version of the food need to comply with these standards in order to be labelled as enriched. The addition of vitamins and minerals to foods in Canada is regulated by provisions which indicate the food to which micronutrients may be added, which micronutrients may be added, and the levels to which they may be added.

### 3.3. Analysis of the Impact of Current Experiences of Calcium Fortification of Wheat Flour

Mandatory calcium fortification of bread and flour currently exists only in the UK where all wheat flour except wholemeal and self-raising flour is required to be fortified with about 235–390 mg calcium carbonate/100 g flour representing 94–156 mg of elemental calcium per 100 g of flour [23]. The UK Bread and Flour regulation from 1998 originates from early post-war regulations and had the objective to restore the calcium lost during milling in order to improve public health. In 2011, all regulations of foods were reviewed with the objective to remove or reduce the burden on the food industry [63]. With this aim the impact of the removal of mandatory fortification was assessed on health outcomes, industry, and the interest of consumers. The review recommended to continue mandatory calcium fortification of flour, as data from the National Diet and Nutrition Survey (NDNS) used to model the impact of removing the mandatory fortification of flour found that the proportion of individuals with Lower Reference Nutrient Intake (LRNI) would increase from 15% to 21% for girls; from 8% to 12% for boys; and from 6% to 9% for women aged 19 to 64 years [64]. Furthermore, except for those aged three years or less, there would be a general downward shift in population intakes of calcium [64]. Survey data showed that the impact of removing mandatory fortification of flour would be greater in low socioeconomic groups as they tend to have lower intakes of these and other nutrients compared with the general population and bread makes a larger contribution to their nutrient intake [63].

Calcium fortification of wheat flour with 200 mg of calcium per 100 g was mandatory in Denmark from 1954 until 1987. Regulations were modified in 1987 and nowadays calcium fortification of wheat flour in Denmark is voluntary. The impact of this change on the Danish population was evaluated in a study conducted in 1998 and concluded that calcium fortification of flour provided 30% of calcium intake and that the cessation of mandatory fortification of flour with calcium increased the number of adults with low calcium intakes from 6% to 22% [65]. 

With the objective to assess if calcium fortification of food would be safe and decrease the proportion of the adult Finish population with a low calcium intake, a simulation study using the national Finnish dietary survey (FINDIET 2002) was carried out [66]. Finland is one of the countries with the highest average calcium intake in the world, 1159 (SD ± 426) in men and 1036 (SD ± 363) in women, and where 17% of the population is lactose intolerant and at risk of low calcium intake. The study found that if all permitted foods were voluntarily fortified with calcium (bread, juice, milk and sour milk, spreads, yogurt, and milk-based desserts) to the maximum possible level, the percentage of individuals with a low calcium intake would decrease from 20.3% to 3.0% in men and from 27.8% to 5.6% in women. However, at the same time the percentage exceeding calcium upper limits (UL) would increase from 0.6% to 12.7% in men and from 0.1% to 3.8% in women and therefore calcium fortification would pose some risk to the population [66]. On the other hand, data shows that in countries where food fortification with calcium is voluntary, not all foods are fortified. In this way the study also evaluated the impact of voluntary calcium fortification in a more realistic scenario using a probability-based model of FINDIET 2002 and found that calcium fortification of foods would lead up to 15.7% of adult men and 23.2% of adult women with a low calcium intake. In this probability-based model, the proportion of individuals with an intake above the UL would be 1.2% in men and 0.7% in women which would be safe [67]. On the contrary, a simulation study in Canada found that although calcium fortification of flour would improve calcium intake of women, it would lead to more than 5% of men exceeding the upper limit [68].

Food fortification has been shown to be an economical way to increase calcium intake, especially if the deficiency is widespread, the target population is hard to reach, and if the technological capacity and distribution networks already exist in the country [27,67]. Fiedler 2008 estimated that in general, the total fortification cost is comprised by 80% of fortificant cost, 8% of marketing and public education, 7% food control and monitoring, and 5% of other program-specific recurrent production costs [22]. However although micronutrients are not expensive, calcium is required in much higher amounts than other micronutrients and little information exists on the cost-effectiveness of calcium fortification, especially in LMIC. 

## 4. Discussion

Food fortification with micronutrients is widely used in many high-income countries and low-, middle or high- income countries. Many of the countries selected in this study have regulation for the micronutrient fortification of staple foods, especially wheat flour and food grade salt, the most common vehicles for food fortification in public health. With regards to calcium fortification, only the UK has mandatory fortification of wheat flour with calcium, however others have the regulation and legislation for grain flour that can be fortified, including rice and wheat and maize flour, and the calcium compounds and calcium levels allowed. These regulations provide the basis to initiate the implementation of a calcium fortification strategy more easily. Besides, many of these countries have industrialized milling of flour which ensures a higher impact for the strategy.

Evaluation in the UK shows that removing mandatory fortification of flour with 94–156 mg of elemental calcium per 100 mg of flour would lead to a general downward shift in population intakes of calcium. In Denmark, a simulation study showed that mandatory fortification of flour with 200 mg/kg provided 30% of the calcium intake and cessation of mandatory fortification led to an increase in the percentage of the adult population with a low calcium intake from 6% to 22%. Finally, in Finland a study evaluating the impact of voluntary calcium fortification in a probability-based model found that calcium fortification of foods would lead to a reduction of the population with low calcium intake from 20.3% to 3.0% in men and from 27.8% to 5.6% in women.

Concerns related to the effects of calcium supplements on other micronutrient absorption are based on short-term studies. One study reports that calcium supplements inhibit iron absorption by 28% to 55% [69]. However, evidence of prolonged calcium supplementation taken at the same time or separate from meals shows no effect on iron status [70,71,72,73,74,75]. In addition, a randomized controlled trial of long term calcium supplementation (1 g of calcium a day) between meals in lactating women at risk of micronutrient deficiency shows no effect on serum iron, zinc, and magnesium concentrations [75].

The increase in hip fractures and myocardial infarct described in some studies performed in older populations with adequate basal calcium intake are still in debate [76,77]. Calcium supplementation randomized controlled trials conducted in younger populations with low basal calcium intake have shown many benefits and report no clear deleterious effects [16]. There is scarce information about high-quality clinical trials evaluating the effect of increasing calcium intake through dietary interventions in populations with low basal calcium intake. The implementation of calcium food fortification strategies should take into consideration population age, basal calcium intake, and food habits to tailor the intervention. These considerations merit assessment in the planning and design stages of any public health intervention at the population level.

Dietary reference values for calcium vary widely depending on the reference guidelines and are under continuous debate. Most were established around ten years ago taking into account calcium needs for bone health. However, new evidence merits reviewing these values. Nevertheless and despite the potential variation in the recommendations, most LMICs do not even reach the lower recommended levels [16]. In addition, any recommendation to introduce a calcium fortification strategy should take into account the basal calcium intake of the population and the gap to achieve the recommended levels, particularly in view of the strong differences in calcium intake between HICs and LICs [1,27]. Considering all the above, we also argue that it would be required to test different doses with simulation studies and assess the impact, effectiveness, and safety of the designed fortification strategy in the selected population [78].

The calcium dietary intake gap between LMICs and HICs seem to be on average around 400–500 mg/day [1,2], which represents a feasible amount to achieve with food fortification [16]. Countries can use as guidelines for their regulation the Codex Alimentarius of the Food and Agriculture Organization and the World Health Organization which are meant to increase the safety and quality of the international food trade and of food consumption worldwide [79]. They provide guidelines that other nations can use as a framework for their own enforceable policies [32]. The Codex suggests that national or regional authorities decide upon the food vehicle for nutrient fortification, taking into consideration the intended purposes of nutrient addition, dietary patterns, socioeconomic situations, and the need to avoid any risks to health [80]. Besides, each country would have to review their policy to comply with regulation of the international trade and economic regulation bodies, for example, the European Union (Regulation No. 1925/2006), The Southern Common Market (MERCOSUR for its Spanish initials), and the Cooperation Council for the Arab States of the Gulf [55,80,81].

## 5. Conclusions

In summary, in view of the benefits of adequate calcium intake at population level and low calcium intake in most LMICs, the strategy to fortified foods with calcium seems promising and feasible as some countries already have regulation and successful experiences of fortification of commonly consumed foods. There are important challenges to fortify some food items, especially flour for baking and bread production, since some changes have been reported to occur that affect organoleptic characteristics and acceptance of the final products. Interference with other micronutrients in the fortification mix as well in total diets could be also drawbacks of the process. National food intake surveys are required regularly to monitor food and nutrient consumption to assess the risks and benefits of fortification programs.

## Figures and Tables

**Table 1 nutrients-12-01022-t001:** Summary of regulations for calcium fortification of staple foods in countries of interest with notes on other fortification regulations.

Country	Regulatory Implementation	Type of Food or Staple Food Regulated	Calcium Fortification Levels
Argentina [38,39]	Voluntary	Any staple food	20%–50% of the RDI representing 200–500 mg of elemental calcium per food serving
Canada [40,41]	Voluntary	Wheat flour	140 mg of elemental calcium per 100 g of flour or 7500 ppm of monocalcium phosphate.
China [42]	Voluntary	Rice, wheat, maize and their products	160–320 mg of elemental calcium per 100 g of staple grains (rice, wheat, and maize and their products)
Denmark [43]	Voluntary	Any processed food	Wheat flour may contain up to 390 mg of elemental calcium per 100 g of flour; water-based non-alcoholic beverages (including water) may contain up to 110 mg of elemental calcium per 100 mL.
Ethiopia [44]	None	None	N/A
India [45]	Voluntary	Any processed food	100% of the RDI representing up to 1000 mg per serving.
Jordan [45,46]	None	None	N/A
Mozambique [47]	None	None	N/A
Qatar [48,49]	Voluntary	Wheat flour	Up to 211.5 mg of elemental calcium per 100 g of flour.
South Africa [50,51]	None	None	N/A
Tanzania [52,53]	None	None	N/A
United Arab Emirates [48,49]	Voluntary	Wheat flour	Up to 211.5 mg of elemental calcium per 100 g of flour.
United States of America [54]	Voluntary	Any processed food except for snack foods	50 mg of elemental calcium per 100 calories of food.
United Kingdom [23,55]	Mandatory	Wheat flour	235–390 mg of calcium carbonate per 100 g of flour, representing about 94–156 mg of elemental calcium per 100 g of flour.
Zambia [56,57]	Voluntary	Wheat and maize flour	Between 111.4 and 144.4 mg of elemental calcium per 100 g of flour for wheat and maize flour.
Zimbabwe [58,59,60]	None	None	N/A

**Table 2 nutrients-12-01022-t002:** Micronutrient fortification standards, and fortification levels of cereal grain in countries of interest, according to the Global Fortification Data Exchange (as of 2018) [62].

Country	Type of Cereal Grain	Fortification Status with Any Nutrient*	Year of Legislation	Percentage of Flour Industrially Milled (**)	Daily Flour Availability (Grams/Capita/Day) (**)	Percentage of Flour Actually Fortified (**)
Argentina	Wheat flour	Mandatory	2002	100%	282	100%
Canada	Wheat flour	Mandatory	1976	100%	233	100%
China	Wheat flour	Voluntary	2012	89%	174	1%
Ethiopia	Wheat flour	Voluntary	2017	55%	86	0%
India	Rice	Voluntary	2018	50%	190	No information
India	Wheat flour	Voluntary	2018	30%	166	No information
Jordan	Wheat Flour	Mandatory	2008	100%	375	90%
Mozambique	Maize flour	Mandatory	2016	30%	149	70%
Mozambique	Wheat flour	Mandatory	2016	100%	40	60%
Qatar	Wheat flour	Voluntary	2015	100%	No information	95%
South Africa	Maize flour	Mandatory	1972	75%	274	80%
South Africa	Wheat flour	Mandatory	2003	100%	165	40%
Tanzania	Maize flour	Mandatory	1975	0%	160	0%
Tanzania	Wheat flour	Mandatory	2011	98%	44	54%
United Arab Emirates	Wheat flour	Voluntary	2015	100%	277	95%
United Kingdom	Wheat flour	Mandatory	1940	98%	270	88%
United States of America	Maize flour	Mandatory	1955	100%	34	29%
United States of America	Rice	Mandatory	1958	100%	19	40%
United States of America	Wheat flour	Mandatory	1942	99%	220	92%
Zambia	Maize Flour	Voluntary	2001	35%	325	0%
Zambia	Wheat flour	Voluntary	2001	100%	33	0%
Zimbabwe	Maize flour	Mandatory	1973	40%	257	43%
Zimbabwe	Wheat flour	Mandatory	1973	40%	86	40%

(*) Any nutrient refers to: calcium, folate, iron, niacin, riboflavin, selenium, thiamine, vitamin A, vitamin B6, vitamin B12, vitamin D, and zinc. (**) Global Fortification Data Exchange: GFDx. Global Fortification Data Exchange. https://fortificationdata.org/#top. 2018.

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
