# Peer review of "Regulatory and Policy-Related Aspects of Calcium Fortification of Foods. Implications for Implementing National Strategies of Calcium Fortification"

_nutrients, 2020, doi:10.3390/nu12041022_

Round 1
Reviewer 1 Report
The objective of the manuscript is to summarize and analyze regulatory and policy-related aspects of calcium fortification of staple foods. The authors conclude that the strategy to fortified foods with calcium seems promising and feasible as many countries already have regulation and successful experiences on fortification of staple foods.
The manuscript is well written, clear and the conclusions are consistent with the analysis of data.
The authors must clarify only two points:
-in the materials and methods section is not clear the database that analyzed to obtain the papers and the specific fortification regulations that discussed in the results section.
- in the results sections the authors write: -We selected 15 countries from different regions of the world to describe the regulations of calcium fortification of staple food (Lines 126-127). The selected 15 countries, are the only that have a specific regulation for the calcium fortification of food, or did the authors selected specific countries in different parts of the world?
Author Response
Response to Reviewer 1 Comments
Point 1: -in the materials and methods section is not clear the database that analyzed to obtain the papers and the specific fortification regulations that discussed in the results section.
Response 1: We added the following information on Materials and Methods:
Page 5: “We review published articles, grey literature in governmental sites of each country and international websites such as Food and Agriculture Organization and the World Health Organization.”
Point 2:- in the results sections the authors write: -We selected 15 countries from different regions of the world to describe the regulations of calcium fortification of staple food (Lines 126-127). The selected 15 countries, are the only that have a specific regulation for the calcium fortification of food, or did the authors selected specific countries in different parts of the world?
Response 2:We selected specific countries in different parts of the world. We modified the phrase to clarify this and added to the methodology section.
Line 154: “We included a convenience sample of 15 countries from different regions of the world and described available regulations of staple food fortification with an emphasis on calcium.”

Reviewer 2 Report
Major Comments
General
My main concern with this paper is that the introduction does not offer a strong enough justification for the reason for undertaking this analysis. The paper does make some interesting points and the tables seem logical and useful but the paper needs to be further strengthened with justification and more discussion on the practicalities of fortification. A modelling example for one low income country with a high prevalence of inadequate calcium intake might strengthen the paper. Some cost-benefit analyses specifically for the low income countries would also be beneficial.
Introduction
The authors discuss the benefit of calcium for bone health and reducing hypertension in pregnancy. It would be useful to provide examples what the medical costs are associated with these health issues or quantify the expected gains in incidence of osteoporosis. There is information on this later in the results section for the UK but it would be good to bring some of this into the introduction.
Some information on the range of calcium intakes for the countries at risk or prevalence of inadequate intakes would also be useful in setting the scene.
Table 1
For the column food vehicles many countries have “numerous”. This does not provide much information. It might be useful to have a supplementary table listing what the Numerous foods are for each country. Alternatively depending on how many numerous food groups there are list them in the current Table 1 or in a footnote.
Table 2
For the last three columns of Table 2 provide a reference as to where this information is from in a footnote to the table.
The discussion of Table 2 is difficult to follow. The table is organised by countries alphabetically but this makes comparisons difficult. Some more thought into organising it differently might be helpful such as dividing into mandatory and voluntary or organising by % of food industrially milled. It would also be helpful to describe one row of the Table in detail. For example in the UK 100% of wheat flour is industrially milled, 272g is available per person per day and 88% of food is fortified. I am not sure what the last column means exactly. Is this the percent of foods in the UK containing flour that are fortified?
Discussion
Do the authors have any comment on whether adding calcium as a fortificant might compromise the bioavailability of other added nutrients.
There is no discussion on what level of fortification might be appropriate.
Although the introduction discusses the higher prevalence of inadequate calcium intake in low income countries much of the discussion is centred on high income countries.
Minor Comments
Page 2 Line 40 – “fortified” should this be fortify?
Major Comments
General
My main concern with this paper is that the introduction does not offer a strong enough justification for the reason for undertaking this analysis. The paper does make some interesting points and the tables seem logical and useful but the paper needs to be further strengthened with justification and more discussion on the practicalities of fortification. A modelling example for one low income country with a high prevalence of inadequate calcium intake might strengthen the paper. Some cost-benefit analyses specifically for the low income countries would also be beneficial.
Introduction
The authors discuss the benefit of calcium for bone health and reducing hypertension in pregnancy. It would be useful to provide examples what the medical costs are associated with these health issues or quantify the expected gains in incidence of osteoporosis. There is information on this later in the results section for the UK but it would be good to bring some of this into the introduction.
Some information on the range of calcium intakes for the countries at risk or prevalence of inadequate intakes would also be useful in setting the scene.
Table 1
For the column food vehicles many countries have “numerous”. This does not provide much information. It might be useful to have a supplementary table listing what the Numerous foods are for each country. Alternatively depending on how many numerous food groups there are list them in the current Table 1 or in a footnote.
Table 2
For the last three columns of Table 2 provide a reference as to where this information is from in a footnote to the table.
The discussion of Table 2 is difficult to follow. The table is organised by countries alphabetically but this makes comparisons difficult. Some more thought into organising it differently might be helpful such as dividing into mandatory and voluntary or organising by % of food industrially milled. It would also be helpful to describe one row of the Table in detail. For example in the UK 100% of wheat flour is industrially milled, 272g is available per person per day and 88% of food is fortified. I am not sure what the last column means exactly. Is this the percent of foods in the UK containing flour that are fortified?
Discussion
Do the authors have any comment on whether adding calcium as a fortificant might compromise the bioavailability of other added nutrients.
There is no discussion on what level of fortification might be appropriate.
Although the introduction discusses the higher prevalence of inadequate calcium intake in low income countries much of the discussion is centred on high income countries.
Minor Comments
Page 2 Line 40 – “fortified” should this be fortify?
Author Response
Response to Reviewer 2 Comments
Major Comments
General
My main concern with this paper is that the introduction does not offer a strong enough justification for the reason for undertaking this analysis. The paper does make some interesting points and the tables seem logical and useful but the paper needs to be further strengthened with justification and more discussion on the practicalities of fortification. A modelling example for one low income country with a high prevalence of inadequate calcium intake might strengthen the paper. Some cost-benefit analyses specifically for the low income countries would also be beneficial.
Thank you for your comment. We have added in the introduction some information of global calcium inadequate intakes and available cost benefit analysis, however cost benefit analysis are from high income countries and none was found on low income countries. (see below replies to the different points raised by the reviewer).
Point 1: The authors discuss the benefit of calcium for bone health and reducing hypertension in pregnancy. It would be useful to provide examples what the medical costs are associated with these health issues or quantify the expected gains in incidence of osteoporosis. There is information on this later in the results section for the UK but it would be good to bring some of this into the introduction.
Response 1: Thank you for this comment. We have moved information of cost to the introduction.
Page 3: “An analysis in the UK assessed that if calcium fortification prevented 2% of fractures annually, stopping mandatory calcium fortification would increase social care cost by £3.06 million per year and by £22.39 million per year in the National Health System.1 Sandman 2015 estimated that the annual costs for 200 mg calcium per day was 0.22 Euros per person in 2014 and that the implementation of a vitamin D plus calcium fortification programme in Germany would cost 41 million Euros per year while saving 315 million Euros per year as a result of reduced fracture costs.2”
Point 2: Some information on the range of calcium intakes for the countries at risk or prevalence of inadequate intakes would also be useful in setting the scene.
Response 2: We have included information on calcium intake in different countries and regions of the world.
Page 2: “The risk of inadequate calcium intake is consistently high in many countries of Asia, Africa and Latin America. A review of daily calcium supply available for human consumption showed a very different consumption by region with figures of 474±188 and 639±223 in Africa and Asia, respectively and ,858±234, 982±130, and 936±50 mg capita for the Americas, Europe, and Oceania respectively, representing a calcium deficiency risk of 80±31, 57±36 for Africa and Asia and 29±27, 11±7 and 11±4% for the Americas, Europe and Oceania respectively.(9) A review reporting calcium dietary intake found that the average national dietary calcium intake ranges from 175 to 1233 mg/day (78 studies from 74 countries) however many countries in Asia have an average dietary calcium intake of less than 500 mg/day. Countries in Africa and South America present calcium intakes between about 400 and 700 mg/day, although there were many countries with no information that would probably reduce these averages.(10) A review reporting dietary intakes of pregnant women shows consistently low calcium intakes across Asian, African and Latin American countries (105 studies, 73,958 pregnant women from 37 countries) [24,26]. The mean calcium intake in pregnant women of low- and middle-income countries was 648 mg/day (95% confidence interval (CI) 569–727) whereas that from high-income countries was 948 mg/day (95% CI 872–1024).(11)”
Point 3: For the column food vehicles many countries have “numerous”. This does not provide much information. It might be useful to have a supplementary table listing what the Numerous foods are for each country. Alternatively depending on how many numerous food groups there are list them in the current Table 1 or in a footnote.
Response 3: We have updated the use of numerous to “any staple food,” “all processed staple foods,” or “all processed staple foods except for snack foods” depending on the laws of each country. Where numerous had been used, these new terms are better indicate that all staple foods (or all processed staple foods) may be fortified with calcium in that country. In these countries there is no a unique regulation for each and every food, instead, there are regulations for fortification of staple foods in general.
Point 4: For the last three columns of Table 2 provide a reference as to where this information is from in a footnote to the table.
Response 4: We have included a footnote to show that this information comes from the Global Fortification Data Exchange
“(**) Global Fortification Data Exchange: GFDx. Global Fortification Data Exchange. https://fortificationdata.org/#top. 2018”
Point 5: The discussion of Table 2 is difficult to follow. The table is organised by countries alphabetically but this makes comparisons difficult. Some more thought into organising it differently might be helpful such as dividing into mandatory and voluntary or organising by % of food industrially milled. It would also be helpful to describe one row of the Table in detail. For example in the UK 100% of wheat flour is industrially milled, 272g is available per person per day and 88% of food is fortified. I am not sure what the last column means exactly. Is this the percent of foods in the UK containing flour that are fortified?
Response 5: We have added an example of one of the rows and organized the discussion of the information presented in Table 2.
“3.2. Micronutrient fortification standards and level of staple grains available for human consumption in countries of interest
“The Global Fortification Data Exchange provides information on current legal fortification standards, industrialized food production, food availability for human consumption and percentage of fortified food.(51) The UK is the only country with mandatory fortification of wheat flour with calcium since 1942, that 98% of wheat flour is industrially milled, that 272g of wheat flour is available for consumption per person per day and that 88% of wheat flour currently produced is being fortified (see Table 2). However, other countries have mandatory or voluntary regulations of food fortification with at least one micronutrient. Countries with extant regulations present an opportunity to add other micronutrients such as calcium to current regulations. In our convenience sample of countries regulations date from the year 1995 in the USA to 2018 in India. Information on the percentage of flour industrially milled is important as industrially produced foods facilitate the attainment of mass fortification. For example, some countries such as Ethiopia, India and Mozambique have low percentage consumption of industrialized flour which would impair the implementation and lower the effectiveness of mandatory fortification of staple grains. In our convenience sample of countries the percentage of flour industrially milled ranges from 0% in Tanzania to a 100% in several countries. Ideally, data on dietary micronutrient intake from population nutritional surveys would be necessary to design a fortification strategy, however this information is not always available. On the other hand, Table 2 presents data on daily food availability data that --although not the optimal measurement,-- is available for many countries. Daily food availability data is the amount of food available per individual in each country calculated from supplies through domestic production or imports. Table 2 shows the daily food availability for each of the selected countries and this information can be used to compare amounts of foods available in different countries. In our convenience sample of countries daily food availability ranges from 375 in Jordan to 33 in Zambia. Finally, the reported percentage of food actually fortified provides an estimate of the success in the process of each fortification strategy. Percentage of Food Actually Fortified provides the percentage of food in column 2 of Table 2 that is being fortified for each country. In our convenience sample of countries the percentage of food being fortified ranges from 0% in Ethiopia and Tanzania to 100% in Argentina and Canada. This indicates that for some countries voluntary food fortification achieves 100% fortification whereas in some other countries voluntary food fortification leads to little impact. On the other hand mandatory food fortification does not always ensure 100% of the food being fortified. Voluntary legislations in China, Zambia, Tanzania or Ethiopia show no or little impact on increasing the percentage of some fortified foods (0 to 54%), whereas in Jordan, United Arab Emirates and Qatar voluntary fortification seems to achieve very similar percentages of food fortification than a mandatory fortification strategy, above 90%.”
Point 6: Do the authors have any comment on whether adding calcium as a fortificant might compromise the bioavailability of other added nutrients.
Response 6: The main concern of increasing calcium intake was on iron bioavailability. We have added information that supports that calcium supplementation does not produce an effect of iron status.
Discussion, page 12: “Concerns related to the effects of calcium supplements on other micronutrients absorption are based on short-term studies. One study reports that calcium supplements inhibit iron absorption by 28-55%.(59) However, evidence of prolonged calcium supplementation taken at the same time or separate from meals shows no effect on iron status. (60–65) In addition, a randomized controlled trial of long term calcium supplementation (1 g of calcium a day) between meals in lactating women at risk of micronutrient deficiency shows no effect on serum iron, zinc a magnesium concentrations. 12”
Point 7: There is no discussion on what level of fortification might be appropriate.
Response 7: We have added information on the fortification level in the discussion.
“The calcium dietary intake gap between LMICs and HICs seem to be on average around 400–500 mg/day, although for some countries the gap seems to be higher, (10,66) which represents a feasible amount to achieve with food fortification.(67)”
Point 8: Although the introduction discusses the higher prevalence of inadequate calcium intake in low income countries much of the discussion is centred on high income countries.
Response 8: Most of the information on cost benefit presented in the discussion is from HIC as there is no information from LMIC. However completing the above comments we have incorporated information on the gap that LMIC require to achieve adequate calcium intake.
Discussion page 11: “Besides a randomized controlled trial of long term calcium supplementation between meals in lactating women at risk of micronutrient deficiency shows no effect of 1 g of calcium a day on serum iron, zinc and magnesium concentrations. 12
The calcium dietary intake gap between LMICs and HICs seem to be on average around 400–500 mg/day, although for some countries the gap seems to be higher, (10,66) which represents a feasible amount to achieve with food fortification.(67)”
Minor Comments
Point 9: Page 2 Line 40 – “fortified” should this be fortify?
Response 9: Thank you for this, we have made the correction

Round 2
Reviewer 2 Report
The authors have addressed all of the comments raised in the previous review. I have no further comments to add. Recommend to accept without further revision.
Author Response
Response to Editor:
We are very grateful about these new considerations regarding our manuscript and following such remarks we add the following texts in the manuscript introduction and discussion.
Introduction, page 3: “Considerations regarding the interpretation of the results of studies involving calcium supplements or increasing dietary calcium intake from food sources need to be taken into account in this discussion. Furthermore the value of results coming from randomized controlled trials and observational studies need to be considered. In a recent review we showed that randomized controlled trials have refuted the side effects of increasing calcium intake like detriment of iron status, formation of renal stones and myocardial infarction in older people.(16) However in supplementation studies, some deleterious effects, such as post‑partum bone resorption and gastrointestinal discomfort have been reported.(16)
Adaptation to low calcium intakes has been described in balance studies. (17) However, adaptations to situations of low calcium intake involve a rise in PTH, which lead to an increase of intracellular calcium in vascular smooth muscles and adipocytes resulting in vasoconstriction and lipogenesis.(18,19)”
Discussion, page 13: “Dietary reference values for calcium vary widely depending on the reference guidelines and are under continuous debate. Most were stablished around ten years ago taking into account calcium needs for bone health. However, new evidence merits reviewing these values. Nevertheless and despite the potential variation in the recommendations, most LMICs do not even reach the lower recommended levels.(16) In addition, any recommendation to introduce a calcium fortification strategy should take into account the basal calcium intake of the population and the gap to achieve the recommended levels, particularly in view of the strong differences in calcium intake between HICs and LICs.(3, 69) Considering all the above, we also argue that it would be required to test different doses with simulation studies and assess the impact -effectiveness and safety- of the designed fortification strategy in the selected population.(70)”
